# Steamed Multigrain Bread Prepared from Dough Fermented with Lactic Acid Bacteria and Its Effect on Type 2 Diabetes

**DOI:** 10.3390/foods12122319

**Published:** 2023-06-09

**Authors:** Jiacui Shang, Shuiqi Xie, Shuo Yang, Bofan Duan, Lijun Liu, Xiangchen Meng

**Affiliations:** 1Key Laboratory of Dairy Science, Ministry of Education, Northeast Agricultural University, Harbin 150030, China; 2Food College, Northeast Agricultural University, Harbin 150030, China

**Keywords:** grains, lactic acid bacteria, steamed bread, low glycemic index, type 2 diabetes

## Abstract

Multigrain products can prevent the occurrence of chronic noninfectious diseases such as hyperglycemia and hyperlipidemia. In this study, multigrain dough fermented by lactic acid bacteria (LAB) was used for the preparation of good-quality steamed multigrain bread, and its effects on type 2 diabetes were investigated. The results showed that the multigrain dough fermented with LAB significantly enhanced the specific volume, texture, and nutritional value of the steamed bread. The steamed multigrain bread had a low glycemic index and was found to increase liver glycogen and reduce triglyceride and insulin levels, while improving oral glucose tolerance and blood lipid levels in diabetic mice. The steamed multigrain bread made from dough fermented with LAB had comparable effects on type 2 diabetes to steamed multigrain bread prepared from dough fermented without LAB. In conclusion, multigrain dough fermentation with LAB improved the quality of the steamed bread while preserving its original efficacy. These findings provide a novel approach to the production of functional commercial foods.

## 1. Introduction

Diabetes mellitus is a common endocrine/metabolic condition that is associated with various cardiovascular complications [1]. Type 2 diabetes is the most prevalent manifestation of diabetes, and effective methods for its prevention and treatment are required [2]. A suitable diet can maintain stable blood glucose levels in patients with diabetes, minimize the requirement for medication, and have a long-lasting and positive impact on patient health [3,4]. Diet is thus extremely important for people with diabetes.

Steamed bread made with wheat flour is consumed widely in China. However, it has a high glycemic index (GI) and is thus unsuitable for diabetic patients [5]. Therefore, the replacement of refined wheat flour with coarse grain has become a hot research topic in the field of food science. Multigrain products have low GIs and can help control blood pressure, lower blood sugar, and improve blood lipid levels [6,7,8]. Unrefined oats can lower blood pressure and serum cholesterol levels, prevent heart disease, and control post-prandial blood sugar [9]. Patients with type 2 diabetes who consume unrefined oats may experience a decreased risk of metabolic and cardiovascular diseases [10].

The influence of other plant-based whole foods, such as peas, on reducing the levels of low-density lipoprotein cholesterol, and thus cardiovascular health, has also been investigated [11]. As β-glucan retards starch digestion, meals high in β-glucan can maintain a low GI [12]. Additionally, the consumption of foods high in β-glucan assists with reducing cholesterol levels. The downside of multigrain food is that various anti-nutritional components and the lack of gluten protein may seriously affect both the nutritional and sensory quality [13,14]. Commercial steamed multigrain bread contains wheat flour and several raw grains as its main ingredients, and it is relatively rare to find steamed multigrain bread made without wheat flour due to its inferior sensory quality.

Lactic acid bacteria (LAB) are probiotic microorganisms that are used as starters [15]. LAB starters can increase the nutritional quality of cereals by reducing the levels of anti-nutritional factors [16,17,18]. For instance, the contents of protein digestibility and free amino acids in the dough were found to increase on the addition of LAB to chickpea dough for fermentation [19]. The presence of pre-fermented wheat bran also has a positive impact on the overall characteristics of the bread [20]. Therefore, LAB starters can be used to improve the quality of grain products [21].

While there has been an increase in research on the use of LAB fermentation for multigrain foods, the development of multigrain dough with LAB fermentation for preparing steamed bread has not reached its full potential. In this study, we used multigrain dough fermented by LAB to prepare steamed bread dough and conducted a preliminary evaluation of the textural characteristics and nutritional quality of steamed multigrain bread. At the same time, the ability of steamed multigrain bread to alleviate type 2 diabetes was explored. The study aimed to improve the quality of steamed multigrain bread and provide a theoretical basis for research into new multigrain products suitable for patients with diabetes.

## 2. Materials and Methods

### 2.1. Preparation of Steamed Bread

The LAB starters RP80 and YO-Prox-700 (Pulse International Trade Co., Ltd., Tianjin, China) were mixed with 100 g of multigrain flour (unrefined oat flour, 30.5%; pea flour, 30.5%; wheat gluten, 35%; oatmeal β-glucan, 4%), followed by the addition of 100 mL of water. The mixture was fermented for 24 h at 37 °C and a relative humidity of 85%. Activated Angel yeast (1%, based on flour weight) and 2% baking soda were then added, and the dough was fermented for 30 min. The dough was divided into 50 g portions, shaped, and proofed for further 20 min at 37 °C and 85% relative humidity. The dough was then placed in a steamer over boiling water and steamed for 25 min to produce steamed bread. The steamed multigrain bread prepared from fermented dough with LAB (FMSB) was then allowed to cool for 1 h at room temperature before analysis. Steamed multigrain breads prepared from the fermented dough without LAB (MSB) and steamed wheat-flour bread (WSB) were used as the blank control and positive control samples, respectively.

### 2.2. Determination of Specific Volume

After steaming, the steamed bread was allowed to cool for 1 h at ambient temperature. The millet displacement method was used to measure the volume of the bread, and the volume (mL) to weight (g) ratio was then used to determine the specific volume [22].

### 2.3. Texture Analysis

The textural properties of the steamed bread, including the specific volume, hardness, gumminess, springiness, chewiness, and resilience, were assessed using a TA.XT-plus texture analyzer (Stable Micro System Co., Ltd., Surrey, UK), as described with some modifications [23]. The steamed bread samples were cut into 15 mm thick slices for assessment. The testing parameters were as follows: cylindrical probe, 35 mm in diameter (P/35); pre-test speed, 1 mm/s; testing speed, 1 mm/s; post-test speed, 1 mm/s; compressed twice with 50% compression; time interval between two compression processes, 5 s.

### 2.4. Determination of Chemical Composition

The enzymatic hydrolysis method developed by McCleary et al. [24] was used to determine the total starch content. Protein was analyzed by the AOAC 950.36 method. Dietary fiber was analyzed using the enzymatic hydrolysis method included in the AOAC 2011.25.

### 2.5. In Vitro Protein Digestion

Protein digestibility was measured using a protocol described by Li et al. [25] with some modifications. Lyophilized steamed bread was crushed and passed through a 60-mesh sieve. The powder (1 g) was incubated in 15 mL of 20 mg/mL pepsin solution at 37 °C for 1 h with shaking. The pH of the samples was adjusted to pH 7.0 using 0.2 mol/L NaOH, after which 15 mL of 5 mg/mL trypsin solution was added and incubated with shaking at 37 °C for 1 h. Lastly, 5 mL of trichloroacetic acid solution was added, mixed, and allowed to stand for 1 h. The samples were then centrifuged (10,000× *g*, 15 min), and the supernatants were retained for further analysis. A BCA protein assay kit (Beyotime Biotechnology, Shanghai, China) was used for assessing the protein content within of the steamed bread samples.

### 2.6. In Vitro Expected Glycemic Response

Starch digestion was measured in vitro as previously described with some modifications [26]. Each sample (100 mg) was combined with 20 mL of sodium acetate buffer and the pH was adjusted to 1.5 with HCl, after which 0.2 mL of pepsin solution (115 U/mL) was added and the mixture was warmed in a water bath at 37 °C for 30 min. The solution was then cooled, and the pH was adjusted to 6.9 with NaOH, after which 1 mL of α-amylase solution (110 U/mL) was added, and the volume was adjusted to 50 mL with phosphate buffer. The mixture was incubated in a 37 °C water bath with shaking at 150 rpm, and enzymes were inactivated by increasing the temperature to 100 °C. The solution was then cooled to room temperature, and 3 mL of sodium acetate buffer (pH 4.5) and 20 μL of glucose amylase (110 U/mL) were added. The solution was incubated in a constant temperature water-bath (55 °C, 45 min) and the glucose content was analyzed using a glucose oxidase assay kit (Nanjing Jiancheng Bioengineering Institute, Nanjing, China).

The areas under the hydrolysis curve of the steamed bread and reference bread (white bread made with wheat) were compared to derive the hydrolysis index (HI). This HI has been found to be strongly correlated with the glycemic index in vivo and is a reliable predictor of the glycemic response to food consumption. The formula eGI = 8.198 + 0.862 HI was used to obtain the expected glycemic index (eGI) [27].

### 2.7. Animals and Experimental Design

Five-week-old male C57BL/6 mice (*n* = 70) were acquired from Beijing Vital River Laboratory Animal Technology Co., Ltd., Beijing, China. The mice were housed in a controlled environment at an ambient temperature of 22 ± 2 °C, humidity of 55% ± 5%, and a 12 h/12 h light/dark cycle with free-access to food and drinking water. Following an acclimatization period of seven days, the mice were randomly divided into two groups. The blank control group (NC, *n* = 15) was fed a basal diet, and the type 2 diabetes model group (*n* = 55) was fed a high-fat diet. After four weeks of feeding, the model mice were intraperitoneally injected with 100 mg/kg body weight of streptozocin (Sigma Aldrich, St. Louis, MO, USA) dissolved in citrate buffer, while the equivalent volume of citrate buffer was injected into the mice in the NC group. A week later, blood samples were collected from the tail vein and the blood glucose levels were assessed using a glucometer (Roche Diagnostics, Mannheim, Germany). Mice having fasting blood glucose levels ≥ 11.1 mmol/L were defined as type 2 diabetic. After model induction, the diabetic mice were randomly assigned to four groups (10 mice/group): the type 2 diabetes model control group (MC); the steamed multigrain bread with LAB group (FM); the steamed multigrain bread without LAB group (MS); and the steamed wheat flour bread as the positive control group (PC). Mice in the FM group were fed diets containing 30% steamed multigrain bread with LAB and 70% basal diet while those in the MS group were fed with 30% steamed multigrain bread without LAB and 70% basal diet, and the PC group was fed with 30% steamed wheat flour bread and 70% basal diet. Feeding continued for five weeks. The NC and MC groups received a basal diet. Body weights, food consumption, and fasting blood glucose levels were monitored weekly after overnight fasting during the five weeks. After the experimental period, all mice were euthanized by anesthesia. Blood samples were collected and centrifuged at 1400× *g* for 10 min, and the sera were stored at −80 °C for further analyses [28]. All animal experiments were approved by the Animal Experimentation Ethics Committee of the Northeast Agricultural University (NEAUEC20220346).

### 2.8. Post-Prandial Blood Glucose

The method of Kawai et al. [29], with slight modifications, was used. Briefly, the MSB, FMSB, and WSB samples were homogenized in water to a final concentration of 200 mg/mL. These suspensions were then orally administered at doses of 2 g/kg body-weight to healthy mice. Blood samples were collected from the tail vein after 30, 60, 90, and 120 min, and the blood glucose levels were assessed using a Roche glucometer. The linear trapezoid method [30] was used to determine the area under the curve (AUC).

### 2.9. Oral Glucose Tolerance Test (OGTT)

The OGTTs were conducted after the experimental period. The mice received oral doses of glucose (2 g/kg body-weight) after a 12-h fast. After 0, 30, 60, 90, and 120 min, blood samples were taken from the tail vein and glucose levels were measured. The linear trapezoid method was used to determine the AUC [30].

### 2.10. Biochemical Analyses

The serum insulin level was determined using an ELISA kit (Beijing Chenglin Bioengineering Institute, Beijing, China) in accordance with the provided protocols. The levels of triglyceride (TG), high-density lipoprotein cholesterol (HDL-C), and low-density lipoprotein cholesterol (LDL-C) in the serum, and glycogen and TG levels in the liver were measured using assay kits (Nanjing Jiancheng Bioengineering Institute, Nanjing, China) following the manufacturer’s instructions.

### 2.11. Statistical Analysis

Date were compared by one-way analysis of variance (ANOVA), followed by Duncan’s multiple range test using SPSS statistical software. *p*-levels < 0.05 were considered statistically significant.

## 3. Results

### 3.1. Specific Volume and Texture of the Steamed Bread

The specific volume and texture are significant indicators of the quality of steamed bread and are critical criteria for product selection by consumers. Compared with MSB, the specific volume of FMSB was significantly higher (Figure 1A, *p* < 0.05). Steamed bread prepared with fermented dough with LAB had significantly reduced hardness and chewiness compared with MSB, although no significant differences were observed between the two samples in terms of elasticity, cohesion, and resilience (Figure 1B–F, *p* > 0.05). Compared with the MSB, the pores of the FMSB were more homogeneous and finer, with a closer resemblance to the WSB. In addition, the color of the FMSB was improved (Appendix A).

### 3.2. Chemical Composition and Protein Digestibility of Steamed Bread

The total starch contents of both FMSB and MSB were markedly lower than those in the WSB sample, while the protein contents were significantly higher (*p* < 0.05); however, these parameters did not differ significantly between the FMSB and MSB (Figure 2A,B). Protein digestibility was significantly higher in the FMSB than in the MSB (Figure 2C, *p* < 0.05). Both MSB and FMSB had significantly higher soluble dietary fiber, insoluble dietary fiber, and total dietary fiber levels than WSB (Figure 2D–F). The amounts of insoluble dietary fiber and total dietary fiber were significantly lower in FMSB compared with MSB (*p* < 0.05).

### 3.3. Predicted Glycemic Response and Post-Prandial Glucose Levels

As shown in Figure 3A, the eGI of steamed bread made with wheat was significantly higher (*p* < 0.05) than that of MSB and FMSB; the eGI values of MSB and FMSB were 49.69 and 50.58, respectively. eGI values below 55 are consistent with the definition of low-GI food. The eGI level of the MSB was comparable to that of the FMSB (*p* > 0.05).

Figure 3B shows the blood glucose levels of healthy mice after feeding steamed bread samples. In the WSB group, the blood glucose levels peaked 30 min after feeding and then steadily declined. Despite the blood glucose levels in the MSB and FMSB groups trending in the same direction as those in the WSB group, the peak value at 30 min was noticeably lower than that of the WSB group. There was no significant difference (*p* > 0.05) between the MSB and FMSB groups in terms of the AUC_glucose_ value (Figure 3C).

### 3.4. Body Weight and Feed Intake

The type 2 diabetes-model mice that received steamed multigrain bread at the end of the trial showed a considerable gain in body weight in comparison to the MC group (Figure 4A). Feed intake in the NC group was markedly lower (*p* < 0.05) than that of the other groups (Figure 4B), while feed intake in the MC group was substantially higher (*p* < 0.05) in comparison to other groups. However, there were no major variations between the PC, MS, and FM groups in the overall amounts of food consumed.

### 3.5. Fasting Blood Glucose

As shown in Table 1, the blood glucose levels in the MC group were markedly higher than those in the NC group. Compared with the MC and PC groups, the blood glucose levels in the MS and FM groups were significantly reduced by week 9. While there was no significant difference between the MS and FM groups, the levels had not returned to normal by the end of the experiment. During the experiment, the blood glucose levels of the MC group were comparable to those of the PC group.

### 3.6. Oral Glucose Tolerance

The glucose tolerance test is commonly used in metabolic research to study glucose metabolism. The OGTT values peaked at 30 min after glucose administration in all groups, which returned to basal levels after 30 min in the NC group while requiring 90 min in the remaining groups to recover (Figure 5A). Data analysis showed that the AUC_glucose_ value for the NC group was markedly lower than that for all other groups (Figure 5B, *p* < 0.05). The AUC_glucose_ values of the MC and PC groups were significantly higher than those of the MS and FM groups, with no significant difference between MS and FM groups (*p* > 0.05).

### 3.7. Serum Insulin Levels

Figure 6A shows that insulin levels in the model group were markedly higher (*p* < 0.01) than those in normal animals at the end of the experimental period. When the diabetic mice were fed FM and MS diets for five weeks, there was a significant decrease in the insulin levels in the FM and MS groups compared with those in the model group (*p* < 0.05), although no major variations were seen between the FM and MS groups.

### 3.8. Blood Lipid Levels

Besides a disordered blood glucose metabolism, diabetes is also characterized by disordered lipid metabolism. As shown in Figure 6B–D, the LDL-C, TG, and HDL-C levels in the MC group were markedly worse than those in the NC group (*p* < 0.05). After receiving a diet of steamed bread for five weeks, the levels of LDL-C, TG and HDL-C in the FM and MS groups recovered to different degrees (*p* < 0.05), and the effects of diet in the FM and MS groups were considerably greater than those in the PC group (*p* < 0.05). No significant differences were seen between the FM and MS groups.

### 3.9. Liver Glycogen and TG Levels

The levels of triglycerides and glycogen in the liver are important indicators of the progression of diabetes mellitus. The results showed that the TG levels in the MC group were significantly higher than those in the NC group while the levels in the FM and MS groups were markedly lower than those in the MC group (Figure 6E, *p* < 0.05). The liver glycogen level in the MC group was significantly lower than that in the NC group (Figure 6F, *p* < 0.05). The liver glycogen contents in the FM and MS groups were significantly higher than those in the MC group. In addition, the liver TG and glycogen levels in the FM group were comparable to those in the MS group.

## 4. Discussion

LAB are frequently employed as starter cultures for the fermentation of a wide range of foods, and they can increase the fermented foods’ nutritional value, flavor, and shelf life [31,32,33]. In the present study, we used dough fermented with LAB to prepare steamed multigrain bread and found that the volume, hardness, and chewability of steamed multigrain bread were improved, consistent with the results of Xing et al. [21]. LAB fermentation enables the degradation of proteins and fiber, improves the ductility of the dough, and leads to a denser structure of the gluten network, which is conducive to the formation of gluten-protein disulfide bonds. Previous studies have also found that fermentation of quinoa dough with *Lactobacillus* can produce quinoa bread with uniform porosity, good sensory quality, and high nutritional value [34]. In the present study, the pores of the steamed multigrain bread prepared from dough fermented with LAB were more uniform and smaller, resembling those of steamed bread made from wheat flour, with improved color, demonstrating the beneficial effect of LAB on steamed multigrain bread.

LAB produce enzymes that enhance the breakdown of insoluble dietary fibers. This may account for the reduction in the insoluble dietary fiber content of steamed multigrain bread prepared from dough fermented with LAB, as observed in this study. As dietary fibers, lipids, starch, and protein in the multigrain dough form a “protein–sugar–oil” membrane structure; the contact area between protein and protease is reduced, thus reducing the in vitro digestibility of protein in the steamed multigrain bread [35]. During the fermentation process, proteins are partially hydrolyzed, making them more vulnerable to proteases [19], and thus improving the protein digestibility of steamed multigrain bread.

Food starch usually forms intricate matrices with proteins and lipids, leading to variations in GI and intrinsic changes in post-prandial glycemic reactions in humans [36]. The GI of oat starch treated with high molecular weight β-glucan has been found to be reduced, suggesting that β-glucan may reduce the digestibility of starch [37]. Moreover, LAB fermentation did not affect the GI value of steamed multigrain bread with low GI, suggesting that the steamed multigrain bread prepared from dough fermented with LAB was suitable for diabetic patients.

The steamed multigrain bread elicited lower post-prandial glycemic responses than steamed bread made with wheat flour, suggesting that the steamed multigrain bread stabilized the post-prandial blood sugar levels. The low-GI steamed multigrain bread contained large amounts of dietary fiber and protein; the protein forms a protective coating around the starch, thus reducing the contact area between amylase and starch and prolonging gastric emptying [38,39]. Lan et al. [40] found that the post-prandial blood glucose level after eating bread made only from buckwheat flour was lower than that made from wheat flour, which is consistent with our results.

As multigrain products with low GI values maintain and stabilize blood sugar levels, we fed steamed multigrain bread to diabetic mice to explore its effects on type 2 diabetes. The improvement in the nutritional quality of the steamed multigrain bread prepared from dough fermented with LAB assisted with weight maintenance in the mice. Insulin resistance increases the risk of developing type 2 diabetes [41]. The OGTT is a recognized tool used in the diagnosis of type 2 diabetes that is frequently used to measure insulin resistance [42]. In this study, the glucose intolerance of the diabetic mice was improved after feeding with steamed multigrain bread prepared from dough fermented with LAB. In addition, insulin is a crucial hormone that is responsible for the regulation of glucose metabolism and the maintenance of a balanced blood glucose level. It has been reported that reductions in insulin levels and AUC_glucose_ values are related to improvements in insulin resistance. Therefore, the results showed that steamed multigrain bread prepared from dough fermented with LAB could ameliorate insulin resistance in mice with type 2 diabetes.

The liver plays a vital role in fat generation, gluconeogenesis, glycogen accumulation and cholesterol metabolism [43]. It has been reported that increasing the glycogen content and reducing the fat content in the liver are of great significance for alleviating type 2 diabetes [44]. In this study, the liver glycogen levels of the mice were found to be increased, while the TG level was reduced after administration of steamed multigrain bread prepared from dough fermented with LAB, indicating that this steamed bread can improve liver metabolism in diabetic mice. Moreover, previous studies have shown that whole-grain brown rice and whole-grain barley can effectively inhibit liver glycogenosis, increase liver glycogen storage, and improve insulin resistance in type 2 diabetes [45,46].

Disordered lipid metabolism is one of the pathological indicators of type 2 diabetes, and it also represents a leading cause of cardiovascular disease in patients with diabetes [47,48]. The blood lipid levels in the steamed multigrain bread groups recovered slightly, with the most marked improvement seen in the LDL-C levels in the group fed with steamed multigrain bread prepared from dough fermented with LAB. The results of a study on 345 people to determine the impact of the processed β-glucan supplements on LDL-C showed that the administration of 3 g of high-viscosity β-glucan induced a 5.5% decrease in LDL-C compared to placebo (wheat fiber) [49]. In addition, studies have shown that the products of LAB fermentation can increase the levels of blood HDL-C in mice [50].

## 5. Conclusions

LAB starters are effective for the improvement of steamed multigrain bread made without wheat flour, resulting in increased specific volume, a softer crumb, and improved crumb browning. These improvements all increase the consumer acceptability of the product. Furthermore, it was found that the effects of a diet containing low-GI steamed multigrain bread on diabetic mice were superior to those of steamed bread made with wheat flour in terms of insulin levels and oral glucose tolerance. These results suggest that this steamed bread is suitable for people with diabetes and that it is effective for stabilizing blood glucose levels. The results of this enhance our understanding of changes in the nutritional quality of cereals induced by LAB fermentation and the protective effects of multigrain bread on diabetic mice.

## Figures and Tables

**Figure 1 foods-12-02319-f001:**
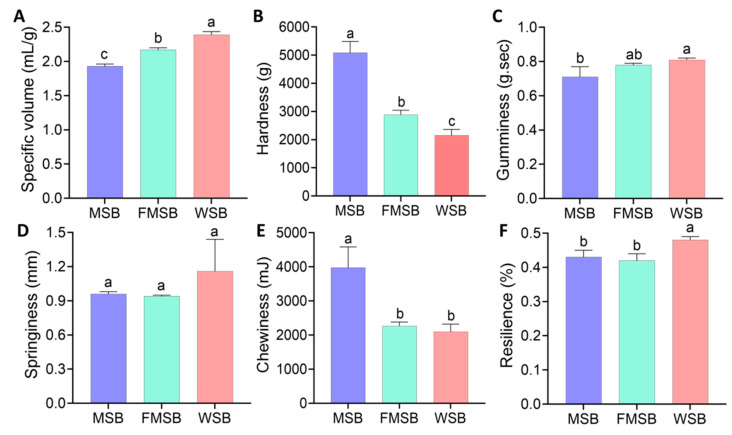
Characteristics of the steamed bread: (**A**) specific volume; (**B**) hardness; (**C**) gumminess; (**D**) springiness; (**E**) chewiness; (**F**) resilience. MSB, steamed multigrain bread without lactic acid bacteria; FMSB, steamed multigrain bread with lactic acid bacteria; WSB, steamed wheat flour bread. Different letters represent significant differences (*p* < 0.05) between groups.

**Figure 2 foods-12-02319-f002:**
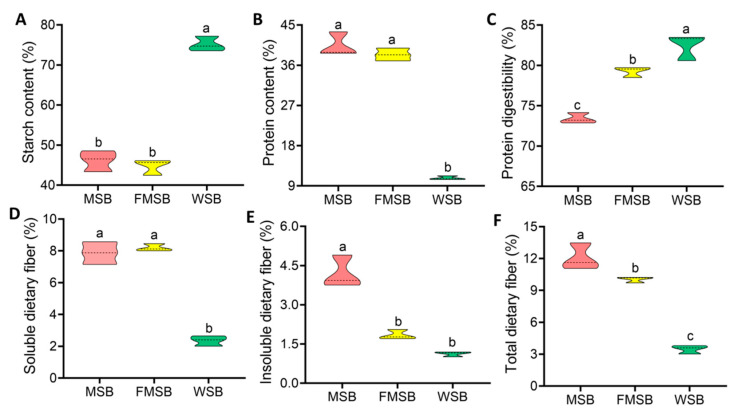
Nutritional content of the steamed bread: (**A**) starch content; (**B**) protein content; (**C**) protein digestibility; (**D**) soluble dietary fiber content; (**E**) insoluble dietary fiber content; (**F**) total dietary fiber content. MSB, steamed multigrain bread without lactic acid bacteria; FMSB, steamed multigrain bread with lactic acid bacteria; WSB, steamed wheat flour bread. Dotted line represents median (50th percentile). Different letters represent significant differences (*p* < 0.05) between groups.

**Figure 3 foods-12-02319-f003:**
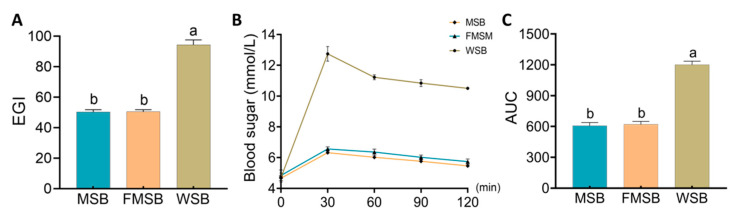
Expected glycemic index of the steamed bread and post-prandial blood glucose in vivo: (**A**) expected glycemic index (EGI); (**B**) post-prandial blood glucose in vivo; (**C**) areas under the curve for glucose (AUC). MSB, steamed multigrain bread without lactic acid bacteria; FMSB, steamed multigrain bread with lactic acid bacteria; WSB, steamed wheat flour bread. Different letters represent significant differences (*p* < 0.05) between groups.

**Figure 4 foods-12-02319-f004:**
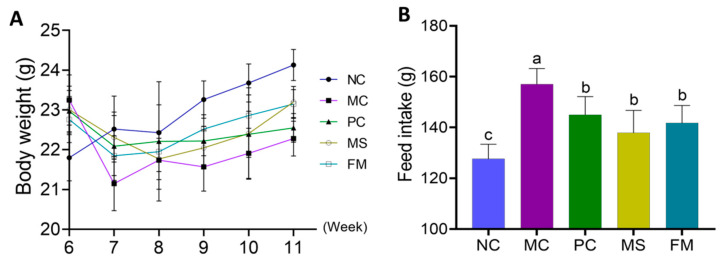
Effects of the steamed bread on body weight and feed intake: (**A**) body weight; (**B**) feed intake. NC, normal control group; MC, model control group; PC, steamed wheat flour bread group; MS, steamed multigrain bread without lactic acid bacteria group; FM, steamed multigrain bread with lactic acid bacteria group. Different letters represent significant differences (*p* < 0.05) between groups.

**Figure 5 foods-12-02319-f005:**
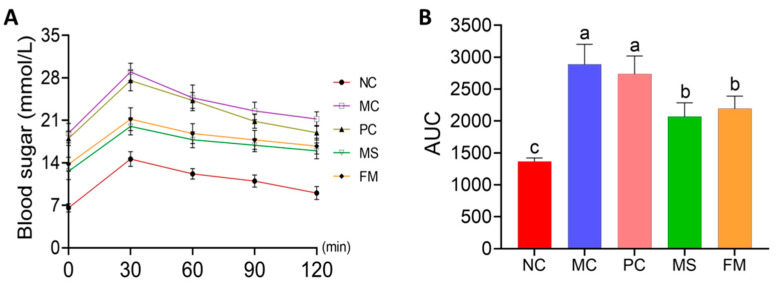
Effects of the steamed bread on blood glucose concentrations and the areas under the curve for glucose in the oral glucose tolerance test: (**A**) blood glucose concentrations; (**B**) areas under the curve for glucose. NC, normal control group; MC, model control group; PC, steamed wheat flour bread group; MS, steamed multigrain bread without lactic acid bacteria group; FM, steamed multigrain bread with lactic acid bacteria group. Different letters represent significant differences (*p* < 0.05) between groups.

**Figure 6 foods-12-02319-f006:**
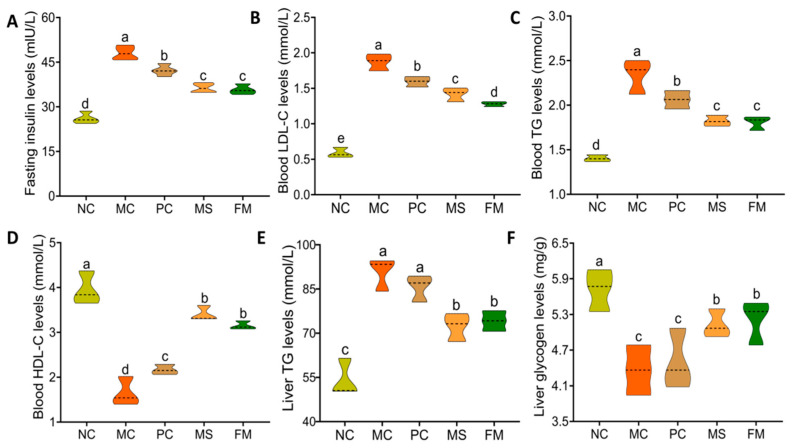
Effects of the steamed bread on the levels of insulin, blood lipids, and liver factors in mice: (**A**) fasting insulin levels; (**B**) blood LDL-C levels; (**C**) blood TG levels; (**D**) blood HDL-C levels; (**E**) liver TG levels; (**F**) liver glycogen levels. NC, normal control group; MC, model control group; PC, steamed wheat flour bread group; MS, steamed multigrain bread without lactic acid bacteria group; FM, steamed multigrain bread with lactic acid bacteria group. Dotted line represents median (50th percentile). Different letters represent significant differences (*p* < 0.05) between groups.

**Table 1 foods-12-02319-t001:** Effects of the steamed bread on fasting blood glucose levels.

Groups	7 Weeks (mmol/L)	8 Weeks (mmol/L)	9 Weeks (mmol/L)	10 Weeks (mmol/L)	11 Weeks (mmol/L)
NC	4.54 ± 0.37 ^b^	4.56 ± 0.20 ^b^	4.54 ± 0.15 ^c^	4.48 ± 0.17 ^c^	4.40 ± 0.06 ^c^
MC	20.88 ± 1.83 ^a^	20.98 ± 0.58 ^a^	20.64 ± 0.49 ^a^	20.56 ± 0.39 ^a^	20.32 ± 0.24 ^a^
PC	21.87 ± 1.10 ^a^	22.31 ± 0.45 ^a^	21.93 ± 0.60 ^a^	21.63 ± 0.53 ^a^	21.20 ± 0.32 ^a^
MS	22.44 ± 0.65 ^a^	22.31 ± 0.57 ^a^	18.77 ± 0.40 ^b^	16.98 ± 0.60 ^b^	16.34 ± 0.73 ^b^
FM	22.12 ± 0.81 ^a^	22.14 ± 0.32 ^a^	18.56 ± 0.80 ^b^	17.44 ± 0.34 ^b^	16.90 ± 0.38 ^b^

Note: NC, normal control group; MC, model control group; PC, steamed wheat flour bread group; MS, steamed multigrain bread without lactic acid bacteria group; FM, steamed multigrain bread with lactic acid bacteria group. Different letters represent significant differences (*p* < 0.05) between groups.

## Data Availability

The data presented in this study are available on request from the corresponding author.

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
