# Peer review of "Steamed Multigrain Bread Prepared from Dough Fermented with Lactic Acid Bacteria and Its Effect on Type 2 Diabetes"

_foods, 2023, doi:10.3390/foods12122319_

Round 1
Reviewer 1 Report
Please provide a short title for the figures: Figure 2, Figure 4, Figure 5 and figure 6.
Please describe in materials and methods how the analyzes of hardness, chewiness, elasticity, cohesion and resilience were carried out.
Fig. 1: Figure captions must be self-explanatory. Please provide information on what the letters A - F in the legend refer to.
Figure3: The caption needs information about the means of “EGI” and “AUC”
Improve figures, especially 3B, 4A and 5A. A bar chart might be better for visualizing differences between tests.
In the discussion: "LAB can activate enzymes in cereals during fermentation", it is worth discussing a little about the enzymes produced by LAB that could be involved in improving the metabolism of insoluble dietary fibers, not necessarily the activation of enzymes in cereals
Please review English with native speaker.
Author Response
1. Please provide a short title for the figures: Figure 2, Figure 4, Figure 5 and figure 6.
AU: Thank you for your constructive suggestion. We have added short titles for the figures. Figure 2. Nutritional content of the steamed bread. Figure 4. Effects of the steamed bread on body weight and food intake. Figure 5. Effects of the steamed bread on blood glucose concentrations and the areas under the curve for glucose in the oral glucose tolerance test. Figure 6. Effects of the steamed bread on the levels of insulin, blood lipids, and liver factors in mice.
2. Please describe in materials and methods how the analyzes of hardness, chewiness, elasticity, cohesion and resilience were carried out.
AU: Thank you for your constructive suggestion. The texture analyzer is highly consistent with human sensory property, and is not interfered by personal factors. It can directly record food-related data. We have added the analyzes of hardness, chewiness, elasticity, cohesion and resilience in materials and methods.
3. Fig. 1: Figure captions must be self-explanatory. Please provide information on what the letters A - F in the legend refer to.
AU: Thank you very much for pointing out this issue. We have provided information on what the letters A - F in the legend refer to. (A) specific volume. (B) hardness. (C) gumminess. (D) springiness. (E) chewiness. (F) resilience.
4. Figure3: The caption needs information about the means of “EGI” and “AUC”
AU: Thank you very much for pointing out this issue. We have added the means of “EGI” and “AUC”. (A) expected glycemic index (EGI). (C) areas under the curve for glucose (AUC).
5. Improve figures, especially 3B, 4A and 5A. A bar chart might be better for visualizing differences between tests.
AU: Thank you very much for pointing out this issue. We think the line chart can more clearly express the trends, and describe their differences in the results section.
6. In the discussion: "LAB can activate enzymes in cereals during fermentation", it is worth discussing a little about the enzymes produced by LAB that could be involved in improving the metabolism of insoluble dietary fibers, not necessarily the activation of enzymes in cereals
AU: Thank you for your constructive suggestion. We have revised it. "LAB produce enzymes that enhance the breakdown of insoluble dietary fibers."
7. Comments on the Quality of English Language
Please review English with native speaker.
AU: Thank you very much for pointing out this issue. The manuscript has been proofread.
Reviewer 2 Report
The manuscript involves a complete work concerning the study of fermented steamed bread, since its preparation, characterisation of physical properties and its effect on the metabolism of diabetic mice. The authors achieved well the information organization and the document is widely interesting. Therefore, the manuscript needs to be improved by carrying out a minor revision.
The comments are the following:
-Section 2.2: The millet displacement method has several variations. Please explain briefly the procedure used for this type of sample.
-Lipids and ashes are missing in the chemical composition. Please include this data.
-Section 2.6: Please explain briefly the procedure to calculate areas under the hydrolysis curve.
-Figure S1 is fine, however, the scale of the images is not equal for all breads. Hence, pores and macrostructure are difficult to be assessed. Please, homogenize the photos (same illumination, same focal distance, and only one piece of bread).
-Please explain the evaluated textural parameters: their definition and why are they important to characterize bread texture?
-Experimental groups: NC, MC, PC, MS, and FM, are confusing. I suggest including a table for describing them.
-In the conclusion Section, please add more details about why steamed multigrain bread on diabetic mice was superior to that of steamed wheat flour bread.
-Only a misspelling was detected: Section 2.1. "The dough was wieghted..." replaced wieghted by weighted.
Only a misspelling was detected: Section 2.1. "The dough was wieghted..." replaced wieghted by weighted. I recommend grammar and spelling revisions.
Author Response
1.-Section 2.2: The millet displacement method has several variations. Please explain briefly the procedure used for this type of sample.
AU: Thank you very much for pointing out this issue. Two round beakers of the same volume were used. The first beaker was filled with millet, which was scraped flat level to the rim of the beaker with a ruler. The steamed bread was placed in the second beaker, which was then filled with millet from the first beaker. The volume of the millet remaining in the first beaker was measured using a measuring cylinder.
2.-Lipids and ashes are missing in the chemical composition. Please include this data.
AU: Thank you very much for pointing out this issue. Lipid levels were not significant difference between each group, so we didn’t show it. Ashes may be not very relevant to our topic, we didn't test it.
3.-Section 2.6: Please explain briefly the procedure to calculate areas under the hydrolysis curve.
AU: Thank you very much for pointing out this issue. The starch hydrolysis curve follows the first order reaction equation, and the area under the hydrolysis curve can be calculated according to the equation:C=c∞ (tf-t0)-(c∞∕k)[1-e-k(tf-t0)]
C∞: reducing sugar concentration at reaction equilibrium; tf: final time (180 min); t0: initial time (0 min); k: first order kinetic constant.
4.-Please explain the evaluated textural parameters: their definition and why are they important to characterize bread texture?
AU: Thank you very much for pointing out this issue. The texture analyzer is highly consistent with human sensory property, and is not interfered by personal factors. It can directly record food-related data. We have added the hardness, chewiness, elasticity, cohesion and resilience in materials and methods.
5.-Experimental groups: NC, MC, PC, MS, and FM, are confusing. I suggest including a table for describing them.
AU: Thank you very much for pointing out this issue. In addition to the text, we show the meaning of the abbreviation in the corresponding figure captions. NC, normal control group; MC, model control group; PC, steamed wheat flour bread group; MS, steamed multigrain bread without lactic acid bacteria group; FM, steamed multigrain bread with lactic acid bacteria group.
6.-In the conclusion Section, please add more details about why steamed multigrain bread on diabetic mice was superior to that of steamed wheat flour bread.
AU: Thank you for your constructive suggestion. We have added it. "it was found that the effects of a diet containing low-GI steamed multigrain bread on diabetic mice were superior to those of steamed bread made with wheat flour in terms of insulin levels and oral glucose tolerance."
7 -Only a misspelling was detected: Section 2.1. "The dough was wieghted..." replaced wieghted by weighted.
AU: Thank you very much for pointing out this issue. We have revised it.
8 Comments on the Quality of English Language
Only a misspelling was detected: Section 2.1. "The dough was wieghted..." replaced wieghted by weighted. I recommend grammar and spelling revisions.
AU: Thank you very much for pointing out this issue. The manuscript has been proofread.
Reviewer 3 Report
Please look through the reviewed manuscript for my points for clarification / revision.
Please consider the usage of "groups". In some instances it would be better to say "samples" when talking about the bread.
I think "groups" may be more appropriate when discussing the mice study.

The English usage is generally fine. A few points are highlighted, along with my suggestion.
Author Response
1.Please look through the reviewed manuscript for my points for clarification / revision.Please consider the usage of "groups". In some instances it would be better to say "samples" when talking about the bread. I think "groups" may be more appropriate when discussing the mice study.
AU: Thank you very much for your comments and suggestions on our paper. We have changed "groups" to "samples" when talking about the bread. Based on these comments and suggestions, we have made careful modifications on the manuscript.
2. "The English usage is generally fine. A few points are highlighted, along with my suggestion."
AU:Thank you very much for pointing out this issue. The manuscript has been proofread.